# Core-Competitiveness in Partially Observable Networked Market

Submission Id: 1425

## ABSTRACT

In auction theory, a core is a stable outcome where no subgroup of participants can achieve better results for themselves. Core-competitive auctions aim to generate revenue that is achievable in a core. They are particularly important because they not only generate optimized revenue for the seller, but also provide an efficient and stable environment for participants.

We generalize the design of core-competitive auctions to encompass partially observable networked markets (PONM). Unlike traditional auctions, which often deal with scenarios of limited trading activity, our approach to core-competitive auctions for PONM captures the nature of real-world transaction markets, which is a large linking world for the economic entities and commodities circulate among the entities in the market. Our generalizing the auction market to PONM can much improve the liquidity of the auction, and is especially meaningful for the web economics. Specifically, we quantify the upper and lower bounds of the minimum core revenue in PONM, and further prove that there does not exist any truthful auction for PONM which is efficient and core-competitive. Governed by this impossible result, we identify the criteria that the allocation rule for PONM should meet. Based on these criteria, we propose a new class of auction mechanisms for PONM that is individually rational, incentive-compatible, and core-competitive.

## CCS CONCEPTS

• **Networks** → **Network economics**; • **Theory of computation** → **Social networks**; **Algorithmic mechanism design**.

## KEYWORDS

Web Economics, Market design, Auction design, Core competitiveness, Incentive compatibility

**ACM Reference Format:**
Anonymous Author(s). 2023. Core-Competitiveness in Partially Observable Networked Market. In *Proceedings of The Web Conference 2024 (WWW'24)*. ACM, New York, NY, USA, 11 pages. https://doi.org/XXXXXXX.XXXXXXX

## 1 INTRODUCTION

In various markets, especially in larger ones, there exists an information discrepancy among economic entities. This discrepancy arises from two key factors. Firstly, economic entities often hold exclusive market-related information, such as private *valuations* of commodities or individual *connections* to others. Secondly, the

vast size of the market makes it impossible for economic entities to gain comprehensive knowledge of all market participants and their connections.

Despite having a limited view of the overall market, economic entities in these markets can still exert influence over market transactions by strategically disclosing their private information, including their values and economic connections. For instance, in multi-level marketing or viral marketing [9, 19], each participant possesses a constrained market perception, but they hold the capability to enhance or diminish the effectiveness of marketing efforts by selectively disseminating information to others. In intermediated markets [3, 6, 24], intermediaries can exploit asymmetric information between buyers and sellers to facilitate trades that benefit themselves. Similar scenarios are also observed in social-network-driven auction markets [18, 22, 33], P2P systems [11, 23] and crowdsourcing markets [16, 27, 28], etc.

In this work, we formally define the aforementioned market as a Partially Observable Networked Market (PONM). In PONM, each economic entity "has access to" and "possesses" a segment of the market information. We study auction design for PONM, where all agents, including the seller and potential buyers, can be represented by network nodes, while their connections are edges with weights reflecting the transaction costs between agents. An allocation is represented by a simple path, with the terminal node on the path being the winning buyer and the cumulative weights of the path corresponding to transmission costs.

We design auction mechanisms for PONM which generate optimized revenue, specifically aiming for revenue that is no worse than the *core revenue*. The core revenue denotes the seller's revenue when the auction achieves its core outcome. This outcome guarantees that no subset of losing buyers can deviate to alternative outcomes that would result in higher revenue for the seller. Furthermore, in addition to optimizing the seller's revenue, core outcomes also resolve the problem of envy among bidders [12]. We extend the existing *core-competitive* revenue-optimizing auction to the intricate setting of PONM. Essentially, crafting core-competitive auctions in PONM requires striking a balance between allocation efficiency and maximizing the seller's revenue. To tackle this challenging objective, we begin by proving that there is no auction mechanism in PONM whose outcome falls within the core. Given this negative result, our focus shifts to designing truthful auctions in PONM while still preserving core-competitive properties. We affirmatively answer this question by identifying a class of truthful auction rules that exhibit core-competitiveness for PONM.

## 2 RELATED WORKS

### 2.1 Core-Based Auction Optimization

The core in the context of auctions was initially introduced in [2], where the authors proposed core-selection as a standalone auction design goal. Auctions that select core allocations generate competitive levels of sales revenues and limit buyer incentives in many

aspects, which has gained popularity both in theory and in practice [1, 4, 7, 8, 13]. Within the field of auction research, studies on the core can be broadly classified into two groups. The first category investigates *core-selecting* auctions [7, 8, 13], which focuses on designing practical auctions that generate outcomes within the core. Goeree and Lien [13] proved that any equilibrium outcome in the core is equivalent to the Vickrey outcome. In other words, if the Vickrey outcome is not in the core, then no core-selecting auction exists. This reveals a severe incompatibility between truth-telling and core-selection. Therefore, previous research on core-selecting auctions has mainly concentrated on investigating non-truthful auctions that lead to core outcomes within the reported preferences.

In contrast, the second category of core in auctions emphasizes the importance of truthfulness and explores truthful auction mechanisms whose revenue is competitive against a core outcome [12, 25]. This field of research is commonly referred to as *core-competitive* auction design, where the minimum core revenue acts as the benchmark for revenue. The notion of core-competitiveness was first introduced by Goel et al. [12], in which the authors suggested the use of the minimum core revenue as a competitive benchmark for truthful auctions. They focused on the Text-and-Image advertising setting, where there is an ad slot which can be filled with either a single image ad or $k$ text ads, and designed truthful auctions that are core-competitive. Markakis and Tsikiridis [25] further studied mechanisms for binary single-parameter domains where each bidder's request for some type of service is either accepted or rejected, and designed the first deterministic core-competitive mechanism within the domain. Our work differs from the above work in the sense that we focus on PONM, an emerging distributed market model in which designing auction mechanisms with revenue guarantees poses new challenges, even in the scenario of a single item.

## 2.2 Networked Markets

Regarding the complexity of PONM, its scenarios can exhibit a wide range of diversity. In an extreme scenario, not only are buyers' valuations of the commodity and their connections kept as private information, but also the weights (costs) associated with the edges may remain unobservable. This particular setting is the most complex one, which is far beyond the existing research norms. Therefore, we explore a common setting where the edge weights are fixed and known once the edge is established. Numerous real-world scenarios align with this market model. For instance, in distribution markets like flower markets, the business relationships between suppliers and dealers often remain confidential, while transportation costs for moving commodities between locations are typically well-defined, especially when dealing with established carriers. Another example is inter-domain routing [10], where self-interested routers can strategically select paths for traffic routing and the costs of delivering the traffic along the selected path are also known. In addition, our model choice also encompasses the recently emerging diffusion auction model [15, 22, 32], in which the seller aims to sell items to a set of buyers who are distributed in social networks.

The closest line of research is diffusion auction design, which is initiated by Li et al. [22]. Diffusion auction is a special instance of PONM whose objective is to incentivize buyers already joining in the auction to further diffuse the auction information to other

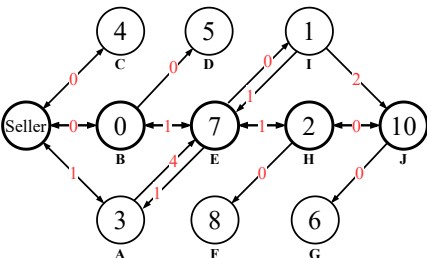

**Figure 1: An example of partially observable networked markets. The left-most circle denotes the seller and all other circles are potential buyers. The number on each directed link represents the transaction costs of using that link.**

buyers via social networks, so as to improve auction outcomes. After [22], many efforts have devoted to diffusion auction design from different angles over the past few years [17, 18, 21, 30, 31, 33]. For recent advances in diffusion auction design, see [15, 32]. Unlike the previous work, we investigate core-competitive auction mechanisms within a more general and realistic model. In our model, the buyers' valuations are mingled with the transaction costs such that any buyer in the market could be critical, which is different from the social network settings where only cut nodes matter. Given the properties highlighted in the previous section, we believe that the core revenue can be taken as a suitable revenue benchmark for diffusion auctions and beyond.

The remainder of the paper is organized as follows. The PONM under investigation, along with the associated auction model and the notion of core-competitiveness, are formally defined in Section 3. Section 4 presents lower and upper bounds for the minimum core revenue. Utilizing these bounds, Section 5 characterizes a class of deterministic auction mechanisms, called deferred allocation auctions, that are proved truthful and core-competitive.

## 3 PRELIMINARIES

In this section, we first introduce the partially observable networked market and the associated auction model, then present the concept of core outcome and core-revenue benchmark.

## 3.1 Partially Observable Networked Market

Consider a seller selling a product/service in a partially observable networked market (PONM), where all agents are connected in a weighted network and can only communicate with her neighbors in the network. Besides the seller, represented by agent 0, the networked market consists of a set of potential buyers, denoted by $N$. Each buyer $i \in N$ has a *private type* $t_i = (v_i, r_i)$, where $v_i$ represents her valuation on the product/service and $r_i$ denotes the neighbors she can communicate with in the market. For each communication link $(i, j)$ with $j \in r_i$, we use $c_{i,j}$ to denote the *transmission cost* of delivering the product/service from $i$ to $j$, which is fixed and known once the communication link is established. A *transaction* in the market is defined by an agent sequence $\{a_i\}_{i=1}^{k}$ with $a_i \in r_{a_{i-1}}$, where $a_0$ and $a_k$ denote the seller and the winning buyer respectively, and $\{a_i\}_{i=1}^{k-1}$ represents the selected path to transmit the commodity. For convenience's sake, we use $\mathcal{G} = (N_0, \{t_i\}, \{c_{i,j}\})$ to define

a partially observable networked market, where $N_0 = N \cup \{0\}$, $\{t_i\} = \{t_i\}_{i \in N_0}$ and $\{c_{i,j}\} = \{c_{i,j}\}_{i \in N_0, j \in r_i}$. Figure 1 demonstrates an example of PONM, where the left-most circle denotes the seller and all other circles are potential buyers. The number in each circle, except the seller, is the buyer's private valuation. The transmission cost is labeled on each edge, and there is an edge $(i, k)$ whenever $k \in r_i$. Given a PONM $\mathcal{G}$, the objective of the seller is to sell the product/service in the whole market $\mathcal{G}$, even though she can only access to a small part of entities in the networked market.

## 3.2 Auction Design in PONM

We model the seller's problem as an auction mechanism design. Formally, denote by $t_i$ buyer $i$'s true type and $\mathbf{t} = (t_i)_{i \in N}$ the type profile of all buyers. For convenience, let $\mathbf{t}_{-i} = \mathbf{t} \setminus \{t_i\}$ be the type profile of all other buyers except $i$. Let $T_i = \mathbb{R}_+ \times \mathbb{P}(N_0)$ be the type space of $i$ where $\mathbb{P}(N_0)$ is the power set of $N_0$, and $T = \times T_{i \in N}$ be the type profile space of all buyers. Since $t_i$ is private information, buyer $i$ can game the mechanism to benefit herself via strategic actions. Accordingly, let $t_i' = (v_i', r_i')$ be $i$'s reported type, where $v_i'$ represents her bid and $r_i'$ is the reported neighbors. As buyer $i$ can only communicate with her neighbors $r_i$, the misreport space of $r_i'$ is limited to $\mathbb{P}(r_i)$. Similarly, let $\mathbf{t}'$ be the reported type profile of all buyers and $\mathbf{t}'_{-i}$ be the reported type profile of all buyers except $i$.

Note that the seller only has access to her neighbors in $\mathcal{G}$, namely $r_0$, at the beginning of the sale, and thus a buyer can participate in the sale only if her neighbors have joined in the sale and further introduced her to the sale. For example, if buyer $E$ does not share the sale information to buyers $H$ and $I$ in Figure 1, then buyers $\{F, G, H, I, J\}$ cannot participate in the sale.

DEFINITION 1. *Given a reported type profile* $\mathbf{t}'$, *we say* $i$ *is a valid buyer if there exists a set of agents* $\{a_j\}_{j=1}^k$ *with* $a_j \in r'_{a_{j-1}}$ *for* $1 < j \leq k$ *and* $i \in r'_{a_k}$.

That is, buyer $i$ is valid if there is a "transaction path" from the seller to $i$ in the reported type profile, and $i$ is invalid if such a "transaction path" does not exist. Given a reported type profile $\mathbf{t}'$, let $V(\mathbf{t}')$ denote all valid buyers. In addition, let $\Pi$ denote the space of all possible transactions with respect to $N_0$ and $\Pi(\mathbf{t}')$ denote the space of transactions given by all valid buyers $V(\mathbf{t}')$. We now formally define the auction mechanisms in PONM.

DEFINITION 2. *An* auction mechanism in PONM *consists of an allocation policy* $\pi : T \to \mathbb{P}(\Pi)$ *and a payment policy* $x = \{x_i : T \to \mathbb{R}\}_{i \in N}$, *and for all reported type profile* $\mathbf{t}'$, $\pi$ *and* $x$ *satisfy the following constraints:*

1) $\pi(\mathbf{t}')$ *and* $x(\mathbf{t}')$ *are independent of* $N \setminus V(\mathbf{t}')$;
2) $\pi(\mathbf{t}') \subseteq \Pi(\mathbf{t}')$ *and* $|\pi(\mathbf{t}')| \leq 1$;
3) $x_i(\mathbf{t}') = 0, \forall i \notin V(\mathbf{t}')$.

Given a reported type profile $\mathbf{t}'$ and an auction mechanism $\mathcal{M}$, $\pi(\mathbf{t}')$ includes the winning buyer and a transmission path to deliver the item, and $x_i(\mathbf{t}')$ denotes the amount each buyer $i$ should pay. We emphasize that although the set of actual bidders would change with the reported type profile, the scenario can be transformed into a direct mechanism design setting: each buyer directly submits her report to the seller; after receiving all reports, the seller executes the mechanism based on *all valid buyers' reports*. The correctness

of such transformation is from the facts that 1) an invalid buyer cannot become a valid buyer by misreporting, and 2) the mechanism is defined on all valid buyers' reports. Therefore, even all buyers reported in the model, but only the buyers who actually participate in the sale will be valid/used.

The *transmission costs* of transaction $\pi(\mathbf{t}')$ is defined as $C(\pi, \mathbf{t}') = \sum_{(i,i+1) \in \pi(\mathbf{t}')} c_{i,i+1}$, where $i, i+1$ are two adjacent buyers in the $\pi(\mathbf{t}')$. To accomplish a transaction $\pi(\mathbf{t}')$, the transmission costs should be covered either by the seller or the buyers. As the seller's revenue comes from the buyers' payments, we can simply ask the seller to immune the transmission costs w.l.o.g. Thus, the seller's revenue (or utility) can be expressed as $R(\mathcal{M}, \mathbf{t}') = \sum_{i \in N} x_i(\mathbf{t}') - C(\pi, \mathbf{t}')$. For each buyer $i \in N$, her utility function is quasi-linear and is defined as follows:

$$u_i(t_i, \mathbf{t}', \mathcal{M}) = z_i(\mathbf{t}')v_i - x_i(\mathbf{t}'). \tag{1}$$

where $z_i(\mathbf{t}')$ is 1 for the winning buyer and 0 otherwise.

We next present two basic properties that an auction mechanism should satisfy. The first property is incentive compatibility. It requires that acting according to their true types forms a dominated strategy for all buyers.

DEFINITION 3. *An auction mechanism* $\mathcal{M}$ *is* incentive-compatible (IC) *if for all* $i \in N$, *all* $t_i$, *and all* $\mathbf{t}'$,

$$u_i(t_i, (t_i, \mathbf{t}'_{-i}), \mathcal{M}) \geq u_i(t_i, (t_i', \mathbf{t}'_{-i}), \mathcal{M}). \tag{2}$$

In any IC auction mechanism, each buyer's utility is maximized by acting truthfully, no matter what the others do. Note that an invalid buyer cannot become a valid buyer by misreporting. As a result, IC constraints are satisfied automatically for all invalid buyers. Another important property is individual rationality, which guarantees that all buyers will receive a non-negative payoff when revealing their true types.

DEFINITION 4. *An auction mechanism* $\mathcal{M}$ *is* individually rational (IR) *if for all* $i \in N$, *all* $t_i$, *and all* $\mathbf{t}'_{-i}$,

$$u_i(t_i, (t_i, \mathbf{t}'_{-i}), \mathcal{M}) \geq 0. \tag{3}$$

The IR property, aka the participation constraint, ensures that all buyers are willing to stay in the auction.

Given a reported type profile $\mathbf{t}'$, let $SW(\pi, \mathbf{t}') = v_w' - C(\pi, \mathbf{t}')$ denote the *social welfare* obtained in $\pi(\mathbf{t}')$, where $w$ represents the winner. Particularly, we use $\pi^*(\mathbf{t}')$ to denote the transaction with the maximum social welfare, and $\pi_i^*(\mathbf{t}')$ to denote the transaction to buyer $i$ with the least transmission costs. It is clear that $\pi^*(\mathbf{t}') \in \arg\max_{\pi_i^*(\mathbf{t}')} SW(\pi_i^*, \mathbf{t}')$. For technique convenience, we treat the seller as a dummy buyer with zero valuation so that $SW(\pi^*, \mathbf{t}') \geq 0$ for all $\mathbf{t}'$. For ease of notation, let $SW^*(\mathbf{t}') = SW(\pi^*, \mathbf{t}')$ and $SW_i^*(\mathbf{t}') = SW(\pi_i^*, \mathbf{t}')$. Accordingly, let $C^*(\mathbf{t}') = C(\pi^*, \mathbf{t}')$ and $C_i^*(\mathbf{t}') = C(\pi_i^*, \mathbf{t}')$ hereafter. We next introduce two properties related to the allocation efficiency of $\pi$.

The first property is called *non-wastefulness*, which requires the mechanism to allocate the commodity whenever possible.

DEFINITION 5. *An auction mechanism* $\mathcal{M}$ *is* non-wasteful (NW) *if* $SW^*(\mathbf{t}') > 0$ *then* $|\pi(\mathbf{t}')| = 1$ *for all* $\mathbf{t}'$.

The second property is *efficiency*, which asks the auction mechanism to allocate the commodity to maximize the social welfare.

DEFINITION 6. *An auction mechanism $\mathcal{M}$ is* efficient *(EF) if $\pi(\mathbf{t}') = \pi^*(\mathbf{t}')$ for all $\mathbf{t}'$.*

Clearly, if an auction mechanism is efficient, it is also non-wasteful, but the reverse is not true. Suppose all buyers in Figure 1 report their types truthfully, then the agent path $\{B, E, H, J\}$ forms the efficient allocation. Besides the allocation efficiency, another desiderata of the seller is revenue. Perhaps the least requirement for revenue is *non-deficit* which asks that $R(\mathcal{M}, \mathbf{t}') \geq 0$ for all $\mathbf{t}'$. An auction mechanism that is non-deficit does not require an injection of funds from the seller. In this work, we consider a more stringer revenue benchmark, which requires the revenue to be competitive against the core outcomes.

## 3.3 Core and Core-Revenue Benchmark

The core is commonly employed as a measure of stability and fairness in coalitional games, which represents a way of distributing the utility generated by a group of agents such that no subgroup of agents would want to deviate. In an auction setting, an outcome is said to be in the core if no subgroup of losing buyers can propose an alternative higher revenue outcome to the seller. Core outcomes not only ensures a revenue guarantee for the seller but also encapsulates some notion of envy–the envy of a set of losing buyers for a set of winning buyers that they can replace [12]. Before introducing the core definition, we first define the coalitional value function or the characteristic function in PONM, which specifies the utilities generated by any group of agents. Given a PONM $\mathcal{G} = (N_0, \{t_i\}, \{c_{i,j}\})$ and a set $S \subseteq N_0$, we define the coalitional value function as

$$W(S) = \begin{cases} \text{SW}^*(\mathbf{t}_S) & 0 \in S, \\ 0 & 0 \notin S, \end{cases} \qquad (4)$$

where $\mathbf{t}_S = \{t_i\}_{i \in S}$ denotes the types reported by $S$.

The pair $(N_0, W)$ defines a coalitional game with transferable utility. The coalition value of a set corresponds to the total utility that can be obtained by the set. Clearly, a coalition that does not contain the seller cannot gain any value. A coalition containing the seller can obtain the maximum utility equal to $\text{SW}^*(\mathbf{t}_S)$–the maximum social welfare achieved in $V(\mathbf{t}_S)$. For any utility profile $\hat{u} = (\hat{u}_i)_{i \in N_0}$, we say $\hat{u}$ is *blocked by* $S$ if there exists a set $S \subseteq N_0$ whose members can be better off by defecting the proposed outcome and redistributing the coalition value among themselves, i.e, $\sum_{i \in S} \hat{u}_i < W(S), \exists S \subseteq N_0$. The *core* is defined as the set of utility profiles that not blocked by any coalition.

DEFINITION 7. *Given a PONM $\mathcal{G}$ and the induced coalitional game $(N_0, W)$, the core, denoted by $Core(N_0, W)$, is defined as the following set of utility profiles:*

$$\left\{ \hat{u} \in \mathbb{R}_+^{|N_0|} : \sum_{i \in N_0} \hat{u}_i = W(N_0), \sum_{i \in S} \hat{u}_i \geq W(S) \quad \forall S \subseteq N_0 \right\}. \quad (5)$$

By definition, any core outcome is efficient, otherwise the grand coalition $N_0$ blocks the outcome. If an outcome $\hat{u}$ is not in the core, then the seller can potentially raise her revenue by negotiating with the losing coalitions. This suggests that the core revenue can be taken as a suitable revenue benchmark against which to compare.

DEFINITION 8. *An auction mechanism $\mathcal{M}$ is* core-competitive *(CC) if for all type profiles $\mathbf{t}$ and the induced coalitional game $(N_0, W)$,*

$$R(\mathcal{M}, \mathbf{t}) \geq CoreRev(N_0, W), \qquad (6)$$

*where $CoreRev(N_0, W) = \min\{\hat{u}_0 | \hat{u} \in Core(N_0, W)\}$ denotes the minimum core revenue.*

In other words, core-competitive auctions ensure the seller a revenue that is at least the minimum core revenue. In the following contents, we investigate auction mechanisms with core-competitive revenues in PONM.

## 4 UPPER/LOWER BOUNDS OF $CoreRev(N_0, W)$

This section analyzes the value of the minimum core revenue $\text{CoreRev}(N_0, W)$. In particular, we identify both lower and upper bounds of $\text{CoreRev}(N_0, W)$. Based on these bounds, in the next section we characterize a set of IC and IR auctions for PONM that are core-competitive.

LEMMA 4.1. *Given any type profile $\mathbf{t}$ and any utility profile $\hat{u} \in Core(N_0, W)$, we have that for all buyers $i \in N$,*

$$0 \leq \hat{u}_i \leq SW^*(\mathbf{t}) - SW^*(\mathbf{t}_{-i}). \qquad (7)$$

PROOF. If $\hat{u}_i < 0$, then $i$ herself forms a blocking coalition. In addition, if $\hat{u}_i > \text{SW}^*(\mathbf{t}) - \text{SW}^*(\mathbf{t}_{-i})$, then $N_0 \setminus \{i\}$ forms a blocking coalition as $W(N_0 \setminus \{i\}) = \text{SW}^*(\mathbf{t}_{-i}) > \text{SW}^*(\mathbf{t}) - \hat{u}_i = \sum_{N_0 \setminus \{i\}} \hat{u}_i$. $\square$

Note that the RHS of (7) is exactly the buyer's utility given in the VCG mechanism [5, 14, 29]. Therefore, Lemma 4.1 suggests that each buyer's utility in the core is no more than her utility given by the VCG mechanism. Based on Lemma 4.1, we next provide a lower bound for the minimum core revenue.

PROPOSITION 1. *Given any type profile $\mathbf{t}$ and the induced coalitional game $(N_0, W)$, we have that*

$$SW^*(\mathbf{t}_{-\pi^*}) \leq CoreRev(N_0, W), \qquad (8)$$

*where $\mathbf{t}_{-\pi^*} = \mathbf{t} \setminus \{t_i\}_{i \in \pi^*(\mathbf{t})}$ denotes the type profile given by all buyers in $N \setminus \pi^*(\mathbf{t})$.*

PROOF. Note that for any buyer $i \notin \pi^*(\mathbf{t})$, the equation $\text{SW}^*(\mathbf{t}) = \text{SW}^*(\mathbf{t}_{-i})$ holds. Thus, based on Lemma 4.1, we know that $\hat{u}_i = 0$ for any utility profile $\hat{u} \in \text{Core}(N_0, W)$ and any buyer $i \notin \pi^*(\mathbf{t})$. As a result, the seller's utility $\hat{u}_0$ can be reformulated as $\sum_{i \in N_0 \setminus \pi^*(\mathbf{t})} \hat{u}_i$ for any $\hat{u} \in \text{Core}(N_0, W)$. According to the definition of $\text{CoreRev}(N_0, W)$, we have that $\sum_{i \in N_0 \setminus \pi^*(\mathbf{t})} \hat{u}_i \geq W(N_0 \setminus \pi^*(\mathbf{t})) = \text{SW}^*(\mathbf{t}_{-\pi^*})$. In other words, the seller's utility in the core is at least $\text{SW}^*(\mathbf{t}_{-\pi^*})$, which implies that $\text{SW}^*(\mathbf{t}_{-\pi^*}) \leq \text{CoreRev}(N_0, W)$. $\square$

Proposition 1 shows that the minimum core revenue is at least the maximum social welfare obtainable after the transaction with the largest social welfare is excluded. Note that $\text{SW}^*(\mathbf{t}_{-\pi^*})$ is essentially the revenue benchmark introduced by Micali and Valiant [26], and therefore Proposition 1 also implies that the core revenue benchmark dominates the Micali-Valiant benchmark for PONM.

Next, we analyze the upper bound of the core revenue benchmark. According to the definition of the core, we know that $\sum_{i \in N_0} \hat{u}_i = W(N_0)$ for any $\hat{u} \in \text{Core}(N_0, W)$, then it is straightforward that

**Algorithm 1:** Critical Buyer Verification

1 **Input:** type profile $\mathbf{t}$, buyer $i$ and efficient transaction $\pi^*(\mathbf{t})$
2 **Output:** *Yes* or *No*
3 identify the winner of $\pi^*(\mathbf{t})$ and denote her as $m$;
4 **while** $r_i \neq \emptyset$ **do**
5     **if** $i \notin \pi^*(\mathbf{t})$ **then return** No;
6     update $\mathbf{t}$ by $((v_i, r_i \setminus \{i+1\}), \mathbf{t}_{-i})$ ;   /* $i+1$ is the next buyer after $i$ in $\pi^*(\mathbf{t})$ */
7     identify the winner in the new efficient transaction $\pi^*(\mathbf{t})$ and denote her as $\tilde{m}$;
8     **if** $\tilde{m} \neq m$ **then return** Yes;
9 **return** No;

$W(N_0)$ is a trivial upper bound of $\mathrm{CoreRev}(N_0, W)$. To obtain a tighter upper bound, we next characterize a set of sufficient conditions for the core based on the concept of critical buyer.

DEFINITION 9. *Given a type profile $\mathbf{t}$, we say $i$ is a critical buyer if there exists $\tilde{r}_i \subseteq r_i$ such that the winners in $\pi^*((v_i, r_i), \mathbf{t}_{-i})$ and $\pi^*((v_i, r_i \setminus \tilde{r}_i), \mathbf{t}_{-i})$ are different.*

That is, a critical buyer is able to change the winner of the efficient allocation by cutting off some of her communication links. For convenience, let $\mathrm{CS}_i^*(\mathbf{t})$ denote the set of such $\tilde{r}_i$ for a critical buyer $i$. Note that the efficient transaction $\pi^*(\mathbf{t})$ is not affected by the communication links of buyers in $N \setminus \pi^*(\mathbf{t})$, so only buyers in $\pi^*(\mathbf{t})$ could become the critical buyers. For example, the efficient transaction in Figure 1 is $\{B, E, H, J\}$. If buyer $B$ does not inform $E$ of the auction information, then the new efficient transaction will be changed to $\{B, D\}$, therefore $B$ is a critical buyer. We can further verify that buyer $E$ is also a critical buyer. However, buyer $H$ is not a critical buyer as no matter how $H$ reports, the winner of the efficient allocation is still buyer $J$. Algorithm 1 presents a simple iterative algorithm to verify whether or not a buyer is critical.

Given any type profile $\mathbf{t}$, for convenience sake we relabel all buyers such that $\pi^*(\mathbf{t}) = \{1, 2, \cdots, m-1, m\}$ where $m$ denotes the winner of $\pi^*(\mathbf{t})$. In addition, we use $N^*(\mathbf{t}) = \{1^*, 2^*, \cdots, (p-1)^*, p^* = m\} \subseteq \pi^*(\mathbf{t})$ to denote the ordered set of all critical buyers $\{i^*\}_{i=1}^{p-1}$ and the buyer $m$, where the label is given by each buyer's position in $\pi^*(\mathbf{t})$. Before characterizing our core conditions, we next present an important property related to $N^*(\mathbf{t})$.

LEMMA 4.2. *For any type profile $\mathbf{t}$, all $i^* \in N^*(\mathbf{t}) \setminus \{m\}$, and all $\tilde{r}_{i^*} \in \mathrm{CS}_{i^*}^*(\mathbf{t})$, the following inequality always holds:*

$$SW^*(\mathbf{t}_{-i^*}) \leq SW^*(\mathbf{t}_{-\tilde{r}_{i^*}}) \leq SW^*(\mathbf{t}_{-i^*+1}), \qquad (9)$$

*where $\mathbf{t}_{-\tilde{r}_{i^*}} = ((v_{i^*}, r_{i^*} \setminus \tilde{r}_{i^*}), \mathbf{t}_{-i^*})$.*

PROOF. Given any type profile $\mathbf{t}$, we know that for any buyer $i^* \in N^*(\mathbf{t})$ there is some $\tilde{r}_{i^*} \subseteq r_{i^*}$ such that $i^*$ can change the winner of $\pi^*(\mathbf{t})$, namely $m$, to another buyer, say $\tilde{m}$, by cutting off $\tilde{r}_{i^*}$. Let $\mathbf{t}_{-\tilde{r}_{i^*}} = ((v_{i^*}, r_{i^*} \setminus \tilde{r}_{i^*}), \mathbf{t}_{-i^*})$ be the updated type profile. According to the definition of $\pi_{i^*}^*(\mathbf{t})$ and $N^*(\mathbf{t})$, for any $i^* \in N^*(\mathbf{t})$ we have that $\pi_{\tilde{m}}^*(\mathbf{t}_{-\tilde{r}_{i^*}}) \cap \{k^*\}_{k=i+1}^p = \emptyset$, i.e., $\pi_{\tilde{m}}^*(\mathbf{t}_{-\tilde{r}_{i^*}}) \subseteq V(\mathbf{t}_{-j^*})$ for any $j^* \in \{k^*\}_{k=i+1}^p$. Since $V(\mathbf{t}_{-i^*}) \subseteq V(\mathbf{t}_{-\tilde{r}_{i^*}})$ and $\pi_{\tilde{m}}^*(\mathbf{t}_{-\tilde{r}_{i^*}}) \subseteq V(\mathbf{t}_{-(i^*+1)})$, the inequality $SW^*(\mathbf{t}_{-i^*+1}) \geq SW^*(\mathbf{t}_{-\tilde{r}_{i^*}}) \geq SW^*(\mathbf{t}_{-i^*})$ holds. □

Based on Lemma 4.2, we are ready to characterize a set of utility profiles in the core.

PROPOSITION 2. *Given any type profile $\mathbf{t}$ and the induced coalitional game $(N_0, W)$, the utility profile $\hat{u}$ is in the core if:*

1) $\sum_{i \in N_0} \hat{u}_i = SW^*(\mathbf{t})$;
2) $0 \leq \hat{u}_{i^*} \leq SW^*(\mathbf{t}_{-i^*+1}) - SW^*(\mathbf{t}_{-i^*}), \forall i^* \in N^*(\mathbf{t})$;
3) $\hat{u}_i = 0, \forall i \in N \setminus N^*(\mathbf{t})$,

*where $SW^*(\mathbf{t}_{-i^*+1})$ is defined as $SW^*(\mathbf{t})$ for $i^* = m$.*

PROOF SKETCH. To prove this proposition, it is sufficient to show that the inequity $\sum_{i \in S} \hat{u}_i \geq W(S)$ holds for all $S \subseteq N_0$. To do so, we first classify all $S$ into several classes based on critical buyers $N^*(\mathbf{t})$. Then, we prove that in each class no group of agents forms a blocking coalition by mathematical induction. The full proof of Proposition 2 is given in the appendix. □

Based on Proposition 2, the following result is straightforward.

COROLLARY 4.3. *Given any type profile $\mathbf{t}$ and any $\hat{u}$ characterized in Proposition 2, the following inequality always holds:*

$$\hat{u}_0 \geq SW^*(\mathbf{t}_{-1^*}). \qquad (10)$$

*In particular, the equality holds by setting $\hat{u}_{i^*} = SW^*(\mathbf{t}_{-i^*+1}) - SW^*(\mathbf{t}_{-i^*})$ for all $i^* \in N^*(\mathbf{t})$.*

Combining the results of Proposition 1 and Corollary 4.3, the following bounds for $\mathrm{CoreRev}(N_0, W)$ are identified.

COROLLARY 4.4. *Given any type profile $\mathbf{t}$ and the induced coalitional game $(N_0, W)$, we have that*

$$SW^*(\mathbf{t}_{-\pi^*}) \leq \mathrm{CoreRev}(N_0, W) \leq SW^*(\mathbf{t}_{-1^*}). \qquad (11)$$

In the example market given in Figure 1, we know that $\pi^*(\mathbf{t}) = \{B, E, H, J\}$ and $N^*(\mathbf{t}) = \{B, E, J\}$. By simple calculation, we have that $SW^*(\mathbf{t}_{-\pi^*}) = SW^*(\mathbf{t}_{-B}) = 4$. Therefore, the minimum core revenue $\mathrm{CoreRev}(N_0, W)$ is exactly 4 according to Corollary 4.4, which is the maximum revenue the seller can obtain by selling the commodity among her direct neighbors $A$, $B$ and $C$.

Given the concept of core-competitiveness and the bounds of $\mathrm{CoreRev}(N_0, W)$ characterized in Corollary 4.4, a natural question is whether it is possible to design auctions that generate outcomes within the core. If so, these auctions are clearly core-competitive. Unfortunately, we next show that there is no auction mechanism in PONM whose outcome is always in the core.

THEOREM 4.5. *No auction mechanism in PONM always generates core outcomes.*

PROOF. It is showed that the *revelation principle* can be extended to networked auction market [17]. Therefore, we can restrict our attention on auctions where all buyers are truth-telling. By definition, any core outcome is efficient, and hence we only need to consider truthful and efficient auctions. Consider a reduced form of PONM, in which all connection weights are zero. For the reduced PONM, Li et al. [20] proved that all truthful and efficient auctions are not weakly budget balanced. That is, the seller can have a lager deficit by implementing the efficient allocation policy in PONM. Since the revenue in the core is non-negative, we conclude that there is no auction in PONM whose outcome is always in the core. □

Theorem [4.5] reveals a severe incompatibility between allocation efficiency and core-competitiveness in PONM, which motivates the following question: Is it possible to design truthful auctions that are core-competitive by sacrificing the allocation efficiency? We provide a positive answer for this question in the next section by characterizing a set of non-wasteful auctions that is individually rational, incentive-compatible and core-competitive.

# 5 DEFERRED ALLOCATION AUCTION

According to Theorem [4.5], we know that designing truthful auctions with revenue guarantee in PONM requires giving up on exact social welfare maximization. To overcome the low revenue issue, this section introduces a family of allocation policies which can provide a flexible tradeoff between allocation efficiency and revenue. Based on the allocation policies, we characterize a set of novel auctions in PONM that are NW, IC, IR and CC. Moreover, we show that this kind of auction produces a core outcome whenever it generates an efficient allocation. Before presenting our main results, we first introduce two important concepts.

DEFINITION 10. *Given buyer $i \in N$, let $\delta_i : T \to \mathbb{P}(N)$ be a link function for $i$, such that given any reported type profile $\mathbf{t}'$, $\delta_i(\mathbf{t}') \subseteq r'_i$ outputs a subset of the reported communication links of buyer $i$.*

Link functions generate a set of communication links tailored for each buyer, which will be used to define the allocation policy. Roughly speaking, given a link function $\delta_i$ of buyer $i$, we will remove a set of links $\delta_i(\mathbf{t}')$ from the market, and then determine whether or not buyer $i$ is qualified to win the commodity by recomputing the efficient allocation. The final winner is pinned by applying the above procedure on a set of ordered buyers. For convenience, let $\delta = (\delta_i)_{i \in N}$ denote the link function profile of all buyers. In addition, let $\mathbf{t}'_{-\delta_i} = ((v'_i, r'_i \setminus \delta_i(\mathbf{t}')), \mathbf{t}'_{-i})$ denote the updated type profile after removing the selected links $\{(i, j)\}_{j \in \delta_i(\mathbf{t}')}$ from $\mathbf{t}'$. Before introducing the allocation policy and the auction mechanism, we next present another concept called critical opponent.

DEFINITION 11. *Given a link function profile $\delta$ and $\mathbf{t}'$, buyer $j \in \pi_i^*(\mathbf{t}')$ is called a critical opponent of $i$ if $\pi_i^*(\mathbf{t}'_{-\delta_j})$ is the transaction with the highest social welfare under $\mathbf{t}'_{-\delta_j}$ and is the unique transaction whose social welfare is no less than that given in $\pi_j^*(\mathbf{t}'_{-\delta_j})$.*

For convenience, let $CO_i^*(\mathbf{t}', \delta)$ denote all $i$'s critical opponents under $\mathbf{t}'$ and $\delta$. Intuitively, if buyer $j$ is a critical opponent of buyer $i$, then $j$ will be the winner in $\pi^*(\mathbf{t}'_{-\delta_j})$ as long as buyer $i$ does not bid. If buyer $i$ eventually wins the item, then she should beat buyer $j$ in $\mathbf{t}'_{-\delta_j}$, otherwise buyer $j$ is more qualified to win. Therefore, buyer $i$'s critical opponents represent her true competitors for winning the commodity, which essentially pin the minimum winning bid of buyer $i$. For example, if buyer $J$ of Figure [1] wins the commodity and $\delta_E(\mathbf{t}') = \{H\}$, then buyer $E$ is a critical opponent of $J$. To see this, note that the transactions with the highest and second highest social welfare in $\mathbf{t}'_{-\delta_E}$ are $\pi_J^*(\mathbf{t}'_{-\delta_E}) = \{B, E, I, J\}$ and $\pi_E^*(\mathbf{t}'_{-\delta_E}) = \{B, E\}$, respectively. If buyer $J$ does not bid, then $E$ will be the winner of $\pi^*(\mathbf{t}'_{-\delta_E})$. To beat $E$ in $\mathbf{t}'_{-\delta_E}$, buyer $J$ should submit a bid of at least $SW_E^*(\mathbf{t}'_{-\delta_E}) + C_J^*(\mathbf{t}'_{-\delta_E}) = 6 + 3 = 9$. Based on the concepts of link function and critical opponent, we now characterize a set of auction mechanisms for PONM in Algorithm [2].

---

**Algorithm 2:** Deferred Allocation Auction (DAA)

1 **Input:** link function profile $\delta$, reported type profile $\mathbf{t}'$
2 **Output:** $(\pi(\mathbf{t}'), x(\mathbf{t}'))$
3 initialize $\pi(\mathbf{t}') = \emptyset$, and $x_i(\mathbf{t}') = 0$ for all $i \in N$;
4 identify the efficient transaction $\pi^*(\mathbf{t}') = \{1, 2, \cdots, m\}$ and compute the associated social welfare $SW^*(\mathbf{t}')$;
5 **return** $(\pi(\mathbf{t}'), x(\mathbf{t}'))$ **if** $SW^*(\mathbf{t}') \leq 0$;
6 **for** $i \leftarrow 1$ **to** $m$ **do**
7     compute $\delta_i(\mathbf{t}')$;
8     **if** $i$ wins in $\pi^*(\mathbf{t}'_{-\delta_i})$ **then**
9         update $\pi(\mathbf{t}')$ by $\pi_i^*(\mathbf{t}'_{-\delta_i})$ and $x_i(\mathbf{t}')$ by $SW^*(\mathbf{t}'_{-i}) + C_i^*(\mathbf{t}'_{-\delta_i})$;
10         identify all $i$'s critical opponents $CO_i^*(\delta, \mathbf{t}')$;
11         **for** $j \in CO_i^*(\delta, \mathbf{t}')$ **do**
12             update $x_i(\mathbf{t}')$ by $\max\{x_i(\mathbf{t}'), SW_j^*(\mathbf{t}'_{-\delta_j}) + C_i^*(\mathbf{t}'_{-\delta_j})\}$;
13         **return** $(\pi(\mathbf{t}'), x(\mathbf{t}'))$;
14     update $x_i(\mathbf{t}')$ by $SW^*(\mathbf{t}'_{-i}) - SW^*(\mathbf{t}'_{-\delta_i})$;
15 **return** $(\pi(\mathbf{t}'), x(\mathbf{t}'))$;

---

Intuitively, DAA adopts a "deferred" allocation policy: It first selects the efficient allocation $\pi^*(\mathbf{t}')$ as the tentative allocation, then determines the qualified winning buyer along $\pi^*(\mathbf{t}')$ based on $\delta$ (*lines 6-13*). In particular, it allocates the commodity to the first buyer $i$ in $\pi^*(\mathbf{t}')$ who wins in $\pi^*(\mathbf{t}'_{-\delta_i})$, and selects $\pi_i^*(\mathbf{t}'_{-\delta_i})$ as the transaction path (*lines 8-9*). According to the definition of $\delta_i$ and the optimality of the shortest path, it is clear that for any $i \in \pi^*(\mathbf{t}')$, we have that $\pi_i^*(\mathbf{t}'_{-\delta_i}) = \pi_i^*(\mathbf{t}') = \{1, 2, \cdots, i\} \subseteq \pi^*(\mathbf{t}') = \{1, 2, \cdots, i, i+1, \cdots, m\}$. Therefore, the final transaction of Algorithm [2] is actually a sub-path of the efficient transaction.

Suppose $w$ is the winner of Algorithm [2], then according to the payment policy, only buyers in $\pi_w^*(\mathbf{t}'_{-\delta_w})$ have non-zero payments. Specifically, for the winner $w$, her payment is pined by her critical opponents, which is defined as

$$\max\left\{SW^*(\mathbf{t}'_{-w}) + C_w^*(\mathbf{t}'_{-\delta_w}), \max_{j \in CO_w^*(\delta, \mathbf{t}')}\{SW_j^*(\mathbf{t}'_{-\delta_j}) + C_w^*(\mathbf{t}'_{-\delta_j})\}\right\}. \tag{12}$$

For each buyer $i \in \pi_w^*(\mathbf{t}'_{-\delta_w}) \setminus \{w\}$, her payment is defined as

$$SW^*(\mathbf{t}'_{-i}) - SW^*(\mathbf{t}'_{-\delta_i}), \tag{13}$$

which is the difference between the maximum social welfare obtained in $\mathbf{t}'_{-i}$ and that obtained in $\mathbf{t}'_{-\delta_i}$. Recall that $\delta_i(\mathbf{t}') \subseteq r'_i$, and therefore $SW^*(\mathbf{t}'_{-i}) - SW^*(\mathbf{t}'_{-\delta_i}) \leq 0$ which means that the seller shall pay $|SW^*(\mathbf{t}'_{-i}) - SW^*(\mathbf{t}'_{-\delta_i})|$ to $i$. This value can be considered as compensation for $i$ for revealing her communication links.

We next present two basic properties that DAA possesses. The first property indicates that DAA is non-wasteful. That is, the commodity can always be allocated whenever $SW^*(\mathbf{t}') > 0$. The second property demonstrates that the critical opponents of the winner exist only when Algorithm [2] produces the efficient transaction.

PROPOSITION 3. *DAA is non-wasteful.*

PROOF. Given any reported type profile $\mathbf{t}'$ and the winner of $\pi^*(\mathbf{t}')$, namely $m$, we know that $\pi_m^*(\mathbf{t}') = \pi_m^*(\mathbf{t}'_{-\delta_m})$. That is, $m$ always wins in $\pi^*(\mathbf{t}'_{-\delta_m})$. According to lines 6-8 of Algorithm 2, the commodity can always be allocated as long as $\mathrm{SW}^*(\mathbf{t}') > 0$. □

PROPOSITION 4. *If there exist critical opponents for the winner of DAA, then the winner must be $m$, i.e, $\mathrm{CO}_w^*(\mathbf{t}') \neq \emptyset \Rightarrow w = m$.*

PROOF. Given any reported type profile $\mathbf{t}'$, suppose $\mathrm{CO}_w^*(\mathbf{t}') \neq \emptyset$ and $w \neq m$. According to Definition 11, we know that for any $i \in \mathrm{CO}_w^*(\mathbf{t}')$, $\mathrm{SW}_i^*(\mathbf{t}'_{-\delta_i})$ is only beaten by $\mathrm{SW}_w^*(\mathbf{t}'_{-\delta_i})$ under $\mathbf{t}'_{-\delta_i}$. Note, however, that if $w \in V(\mathbf{t}'_{-\delta_i})$, then clearly $\{w + 1, \cdots, m\} \subseteq V(\mathbf{t}'_{-\delta_i})$. Based on the optimality of shortest path and the fact that $\mathrm{SW}_m^*(\mathbf{t}') \geq \mathrm{SW}_w^*(\mathbf{t}')$, we know that $v_m' - \sum_{j=w}^{m-1} c_{j,j+1} \geq v_w'$. This leads to the conclusion that $\mathrm{SW}_m^*(\mathbf{t}'_{-\delta_i}) \geq v_m' - \sum_{j=w}^{m-1} c_{j,j+1} - C_w^*(\mathbf{t}'_{-\delta_i}) \geq \mathrm{SW}_w^*(\mathbf{t}'_{-\delta_i})$. Now, $\mathrm{SW}_m^*(\mathbf{t}'_{-\delta_i})$ also beats $\mathrm{SW}_i^*(\mathbf{t}'_{-\delta_i})$ in $\mathbf{t}'_{-\delta_i}$, and thus $i$ cannot be a critical opponent of $w$ according to Definition 11. In sum, if $\mathrm{CO}_w^*(\mathbf{t}') \neq \emptyset$ then $w$ must be $m$. □

According to Algorithm 2, the performance of DAA is entirely determined by the design of $\delta$. On the one hand, the output of $\delta$ determines the difficulty for a buyer to become a winner, or in other words, it determines the allocation efficiency. On the other hand, it also determines the payment amount for each buyer. Intuitively, the seller can balance the allocation efficiency and the revenue by choosing an appropriate $\delta$. In the following contents, we analyze how to design $\delta$ such that the associated DAA is individually rational, incentive-compatible and core-competitive.

## 5.1 Individually Rational and Incentive-Compatible DAA

Recall that the communication links $r_i$ are private information, and buyer $i$ can expose the information of $r_i$ in her own favor. The key purpose of our mechanisms is to incentivize buyers to expose the true information of both valuations and links, and then optimize the seller's revenue to make it core-competitive. In order to incentivize buyers to tell the truth in Algorithm 2, the link function needs to satisfy two properties. The first property is *monotonicity*, which requires the valid buyer set to be non-decreasing with $r_i$.

DEFINITION 12. *For any two type profiles $\mathbf{t} = ((v_i, r_i), \mathbf{t}_{-i})$ and $\tilde{\mathbf{t}} = ((v_i, \tilde{r}_i), \mathbf{t}_{-i})$ such that $i \in \pi^*(\mathbf{t}) \cap \pi^*(\tilde{\mathbf{t}})$, $\delta_i$ is monotonic (MN) if $\tilde{r}_i \subseteq r_i$ then $V(\tilde{\mathbf{t}}_{-\delta_i}) \subseteq V(\mathbf{t}_{-\delta_i})$.*

If $\delta_i$ is monotonic, then reporting all communication links can maximize the set of valid buyers for $\mathbf{t}_{-\delta_i}$, which potentially maximizes the compensation received from the seller according to (13). Therefore, the monotonicity property is align with the buyers' sharing incentives. According to Algorithm 2, only buyers in $\pi^*(\mathbf{t}')$ have opportunities to win the commodity. Thus, in order to improve utilities, the losers in $\pi_m^*(\mathbf{t}) \setminus \pi_w^*(\mathbf{t})$ may behave strategically to influence the output of $\delta_i(\mathbf{t}')$, where $i \in \pi_w^*(\mathbf{t})$, such that Algorithm 2 terminates after $w$. To avoid such kind of behaviors, the link function policy should also be strategy independent, which is formally defined below.

DEFINITION 13. *For any type profile $\mathbf{t}$, and two buyers $i \in \pi_w^*(\mathbf{t})$, $j \in \pi_m^*(\mathbf{t}) \setminus \pi_w^*(\mathbf{t})$, $\delta_i$ is strategy independent (SI) if $\delta_i(\mathbf{t}) = \delta_i(\tilde{\mathbf{t}})$ for all $\tilde{\mathbf{t}} = (t_j', \mathbf{t}_{-j})$ where $i, j \in \pi_m^*(\tilde{\mathbf{t}})$.*

In other words, the strategy independent property requires that for any buyer $i$ before $w$, the output of $\delta_i$ does not vary with the actions of the buyers after $w$. We say $\delta$ is MN/SI if every $\delta_i$ is MN/SI. Next, we show that if $\delta$ satisfies both monotonicity and strategy independence, then the corresponding DAA defined in Algorithm 2 is incentive-compatible and individually rational.

THEOREM 5.1. *If $\delta$ is monotonic and strategy independent, then DAA is individually rational and incentive-compatible.*

PROOF SKETCH. To prove this theorem, we show that no buyer can gain a better revenue by deviating from truthful report, no matter what the other buyers do. For a better illustration, we divide all buyers into four classes: 1) $i \notin \pi^*(\mathbf{t}')$, 2) $i \in \pi_w^*(\mathbf{t}') \setminus \{w\}$ where $w$ denotes the winner in DAA, 3) $i = w$ and 4) $i \in \pi_m^*(\mathbf{t}') \setminus \pi_w^*(\mathbf{t}')$. We prove that if $\delta$ is monotonic and strategy independent, then buyers in each class cannot do better via misreporting. The full proof of Theorem 5.1 is given in the appendix. □

Although DAAs with $\delta$ satisfying MN and SI are individually rational and incentive-compatible, the revenue obtained in such DAAs can be too low. This can be shown by the following example.

*Example 5.2.* Let $\delta^{\emptyset}$ be the link function profile where

$$\delta_i^{\emptyset}(\mathbf{t}') = \emptyset, \forall i \in N, \forall \mathbf{t}' \in T. \tag{14}$$

It is apparently that $\delta^{\emptyset}$ is MN and SI. In addition, the corresponding allocation policy is also efficient. However, adopting such a link function profile can lead to a deficit for the seller. For example, if we apply the DAA with $\delta^{\emptyset}$ in Figure 1, then the final allocation is $\{B, E, H, J\}$. By calculation, we have that $x_B = -4, x_E = -3, x_H = -2, x_J = 8$, and eventually the seller's revenue equals $-3 = -4 - 3 - 2 + 8 - 2$, where the last term $-2$ represents the transmission costs. In fact, the DAA with $\delta^{\emptyset}$ is exactly the VCG mechanism within the PONM setting. Therefore, to design DAAs with competitive revenue, $\delta$ should satisfy additional conditions.

## 5.2 Core-Competitive DAA

This section investigates conditions under which DAA is core-competitive. Recall that the minimum core revenue $\mathrm{CoreRev}(N_0, W)$ is upper bounded by $\mathrm{SW}^*(\mathbf{t}_{-1^*})$ where buyer $1^*$ denotes the first critical buyer in $N^*(\mathbf{t})$. If DAA generates a revenue higher than $\mathrm{SW}^*(\mathbf{t}_{-1^*})$, then clearly it is core-competitive. From Lemma 4.2 and Proposition 2, we know that critical buyers are key to obtain the upper bound. Next, we introduce a condition called *path blocking* (PB) which is interconnected with critical buyers. We prove that if $\delta$ is path blocking, then the corresponding DAA can obtain a competitive revenue that is no less than $\mathrm{SW}^*(\mathbf{t}_{-1^*})$.

DEFINITION 14. *For any type profile $\mathbf{t}$ and $i \in \pi^*(\mathbf{t})$, $\delta_i$ is path blocking (PB) if there is no transaction path from $i$ to $m$ under $\mathbf{t}_{-\delta_i}$.*

In other words, if $\delta_i$ is PB, then under $\mathbf{t}_{-\delta_i}$ all transaction paths from the seller to buyer $m$ do not pass through $i$. We say $\delta$ is PB if all $\delta_i$ is PB. Given any $\delta$ satisfying PB, we can prove the following.

LEMMA 5.3. *If $\delta$ is path blocking, then for all buyers $i$ with $x_i(\mathbf{t}) < 0$, we have that $i \in N^*(\mathbf{t})$ and $\delta_i(\mathbf{t}) \in CS_i^*(\mathbf{t})$.*

PROOF. Given any type profile $\mathbf{t}$, we know that only buyers in $\pi^*_w(\mathbf{t}) \setminus \{w\}$ can have negative payments according to Algorithm 2. For any buyer $i \in \pi^*_w(\mathbf{t}) \setminus \{w\}$, her payment is identical to $\text{SW}^*(\mathbf{t}_{-i}) - \text{SW}^*(\mathbf{t}_{-\delta_i})$. Since $\delta$ is PB, then there is no transaction path from $i$ to $m$ under $\mathbf{t}_{-\delta_i}$, which implies that $\pi^*_m(\mathbf{t}_{-\delta_i}) \subseteq V(\mathbf{t}_{-i})$. Suppose that $m \in \pi^*(\mathbf{t}_{-\delta_i})$, then $m$ must be the winner of $\pi^*(\mathbf{t}_{-\delta_i})$. Since $i \notin \pi^*_m(\mathbf{t}_{-\delta_i})$, then $\text{SW}^*(\mathbf{t}_{-i}) = \text{SW}^*(\mathbf{t}_{-\delta_i}) = \text{SW}^*_m(\mathbf{t}_{-\delta_i})$, resulting in that $x_i(\mathbf{t}) = 0$. Hence, if $x_i(\mathbf{t}) < 0$, then $m \notin \pi^*(\mathbf{t}_{-\delta_i})$, which means that buyer $i \in N^*(\mathbf{t})$ and $\delta_i(\mathbf{t}) \in \text{CS}^*_i(\mathbf{t})$. □

Based on Lemma 4.2 and Lemma 5.3, we next prove that DAA is individually rational, incentive-compatible and core-competitive provided that $\delta$ is MN, SI and PB.

THEOREM 5.4. *If $\delta$ is monotonic, strategy independent and path blocking, then DAA is individually rational, incentive-compatible and core-competitive.*

PROOF. Given any type profile $\mathbf{t}$ and any $\delta$ satisfying MN, SI and PB, let $\text{NP}^*(\mathbf{t}) = \{1^\diamond, \cdots, q^\diamond\} \subseteq \pi^*_w(\mathbf{t}) \setminus \{w\}$ denote the set of buyers with negative payments and let $(q+1)^\diamond$ represent $w$ for convenience sake. Based on Algorithm 2 and Lemma 5.3, the seller's revenue $R(\text{DAA}, \mathbf{t})$ obtained in DAA can be expressed by

$$\sum_{i \in N} x_i(\mathbf{t}) - C^*_w(\mathbf{t}) = \sum_{i \in \text{NP}^*(\mathbf{t})} x_i(\mathbf{t}) + (x_w(\mathbf{t}) - C^*_w(\mathbf{t})). \quad (15)$$

Since $x_w(\mathbf{t})$ is at least $C^*_w(\mathbf{t}) + \text{SW}^*(\mathbf{t}_{-w})$, then $R(\text{DAA}, \mathbf{t})$ is at least

$$\sum_{i \in \text{NP}^*(\mathbf{t})} (\text{SW}^*(\mathbf{t}_{-i}) - \text{SW}^*(\mathbf{t}_{-\delta_i})) + \text{SW}^*(\mathbf{t}_{-w}). \quad (16)$$

According to Lemma 5.3, for all $i \in \text{NP}^*(\mathbf{t})$, we have $\text{NP}^*(\mathbf{t}) \subset N^*(\mathbf{t})$ and $\delta_i(\mathbf{t}) \in \text{CS}^*_i(\mathbf{t})$. Based on Lemma 4.2, the value of (16) is at least

$$\sum_{i=1^\diamond}^{q^\diamond} (\text{SW}^*(\mathbf{t}_{-i}) - \text{SW}^*(\mathbf{t}_{-i+1})) + \text{SW}^*(\mathbf{t}_{-w}) = \text{SW}^*(\mathbf{t}_{-1^\diamond}). \quad (17)$$

Since $1^\diamond \in N^*(\mathbf{t})$, then based on $(15) - (17)$ and Lemma 4.2, we have that $R(\text{DAA}, \mathbf{t}) \geq \text{SW}^*(\mathbf{t}_{-1^\diamond}) \geq \text{SW}^*(\mathbf{t}_{-1^*})$. According to Definition 8 and Corollary 4.4, the DAA is core-competitive. □

According to Algorithm 2, we know that the allocation policy of DAA is not efficient, that is it does not always allocate the commodity to maximize the social welfare. However, our next result shows that the auction outcome must be in the core whenever it produces an efficient allocation.

PROPOSITION 5. *Given a type profile $\mathbf{t}$ and a $\delta$ satisfying MN, SI and PB, if $\pi(\mathbf{t}) = \pi^*(\mathbf{t})$, then $(u_i(\mathbf{t}, \text{DAA}))_{i \in N_0} \in \text{Core}(N_0, W)$.*

PROOF. Since $\delta$ is MN and SI, then all buyers will act truthfully in DAA. Suppose $\pi(\mathbf{t}) = \pi^*(\mathbf{t})$ for a type profile $\mathbf{t}$. According to the payment policy, only buyers in $\pi^*(\mathbf{t})$ could have non-zero utilities. Moreover, based on Lemma 5.3, we have that only buyers in $N^*(\mathbf{t})$ could have positive payoffs. According to (13), for any $i^* \in N^*(\mathbf{t}) \setminus \{m\}$, her payoff $u_{i^*}(\mathbf{t}, \text{DAA})$ equals $\text{SW}^*(\mathbf{t}_{-\delta_{i^*}}) - \text{SW}^*(\mathbf{t}_{-i^*})$. Since $\delta$ is PB, then based on Lemma 5.3 and Lemma 4.2, we have that $u_{i^*}(\mathbf{t}, \text{DAA}) \leq \text{SW}^*(\mathbf{t}_{-i^*+1}) - \text{SW}^*(\mathbf{t}_{-i^*})$, where $i^* + 1 \in N^*(\mathbf{t})$. In addition, according to (12), the winner $m$'s payoff $u_m(\mathbf{t}, \text{DAA})$ is at most $v_m - (\text{SW}^*(\mathbf{t}_{-m}) + C^*_m(\mathbf{t})) = \text{SW}^*(\mathbf{t}) - \text{SW}^*(\mathbf{t}_{-m})$. Based on the core characterizations of Proposition 2, we conclude that the utility profile $(u_i(\mathbf{t}, \text{DAA}))_{i \in N_0} \in \text{Core}(N_0, W)$. □

We next demonstrate two examples to end this section. These examples provide an affirmative answer to the feasibility of designing truthful and core-competitive auctions in PONM.

*Example 5.5.* Let $\delta^r$ be the link function profile where

$$\delta^r_i(\mathbf{t}') = r'_i, \forall i \in N, \mathbf{t}' \in T. \quad (18)$$

According to the definition of $\delta^r$, the valid buyer set $V(\mathbf{t}'_{-\delta^r_i})$ is independent of $r'_i$ and $\delta^r_i(\mathbf{t}')$ is independent of other buyers' reports. Therefore, $\delta^r$ is MN and SI. In addition, as all communication links from $i$ are removed under $\mathbf{t}'_{-\delta^r_i}$, clearly there is no transaction path from $i$ to $m$ in $\mathbf{t}'_{-\delta^r_i}$, i.e., $\delta^r$ is PB. Therefore, the DAA with $\delta^r$ is IC, IR and CC. Applying $\delta^r$ in Figure 1, we get that the final transaction is $\{B, E\}$ and $x_B = 0, x_E = 6$. Eventually, the seller's revenue is $5 = 0 + 6 - 1 > \text{CoreRev}(N_0, W) = 4$. According to (12) and (13), we know that under $\delta^r$, only the winner could have positive payoff. Even though, we emphasize that buyers still have incentives to do the sharing. This is because that spreading the auction information outwardly as far as possible can reduce the competitiveness of other buyers, thus increasing the probability of winning, and can also reduce the number of critical opponents, thus lowering potential payments. To compensate buyers more, thus making them more motivated to share, we can adopt the following link function profile.

*Example 5.6.* We say $j \in r'_i$ is an intermediary of $i$ if there exists $k \neq i$ such that $k \in r'_j$. Let $\delta^I$ be the link function profile where

$$\delta^I_i(\mathbf{t}') = \{j | j \text{ is an intermediary of } i\}, \forall i \in N, \mathbf{t}' \in T. \quad (19)$$

One can further verify that $\delta^I$ also satisfies MN, SI and PB. Given the example market of Figure 1, we have that $\delta^I_B = \{E\}$ and $\delta^I_E = \{I, H\}$. If we apply the DAA with $\delta^I$ in the example market. We can get that the final allocation is $\{B, E\}$ and $x_B = -1, x_E = 6$. Eventually, the seller's revenue is $4 = -1 + 6 - 1 = \text{CoreRev}(N_0, W)$.

## 6 CONCLUSIONS AND FUTURE WORK

This study explored the feasibility of designing auction mechanisms in PONM that generate a revenue competitive against the core revenue. We provided an affirmative answer to this question, by characterizing a class of incentive-compatible auction rules with the core-competitiveness property. Although this work primarily focused on single item settings, the distributed nature of PONM makes the problem of designing core-competitive auctions non trivial, which requires navigating the intricate trade-off between allocation efficiency and revenue.

This research opens up avenues for further investigation and poses several intriguing research questions. One immediate extension is core-competitive auctions in more general auction settings, such as multi-unit auctions or package auctions in PONM. Exploring the dynamics and challenges of designing auctions in these contexts can provide valuable insights into the feasibility and mechanisms for achieving core-competitive outcomes. Additionally, it is also interesting to explore non-truthful auctions in PONM whose outcome is always in the core with respect to the reported types. Investigating such auction mechanisms and analyzing their properties can deepen our understanding of the strategic behavior of participants in PONM auctions and offer alternative approaches to achieving core outcomes.

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

# A  OMITTED PROOF FOR SECTION 4

PROOF OF PROPOSITION 2. To prove this proposition, it is sufficient to show that the inequity $\sum_{i \in S} \hat{u}_i \geq W(S)$ holds for all $S \subseteq N_0$. Given any type profile $\mathbf{t}$, let $\hat{u}$ be any utility profile defined by the three conditions listed in the proposition. According to Lemma 4.2, $SW^*(\mathbf{t}_{-i^*+1}) \geq SW^*(\mathbf{t}_{-i^*})$ for any $i^* \in N^*(\mathbf{t})$, therefore $\hat{u}$ is a feasible utility profile. According to "1)" and "3)", we know that

$$\hat{u}_0 = SW^*(\mathbf{t}) - \sum_{i^* \in N^*(\mathbf{t})} \hat{u}_{i^*}. \tag{20}$$

Based on "2)", we further have that $\sum_{i^* \in N^*(\mathbf{t})} \hat{u}_{i^*}$ is at most

$$\sum_{i^* \in N^*(\mathbf{t})} SW^*(\mathbf{t}_{-i^*+1}) - SW^*(\mathbf{t}_{-i^*}) = SW^*(\mathbf{t}) - SW^*(\mathbf{t}_{-1^*}), \tag{21}$$

where recall that $1^*$ is the first buyer in $N^*(\mathbf{t})$. Therefore, based on (20) and (21), we have that for any $\hat{u}$ defined in the proposition,

$$\hat{u}_0 \geq SW^*(\mathbf{t}_{-1^*}) \geq 0. \tag{22}$$

Since $W(S) = 0$ for any $S$ not including the seller, hence such $S$ cannot be a blocking coalition under $\hat{u}$. Next, consider all $S$ including the seller, we prove that no such $S$ can be a blocking coalition by mathematical induction. For convenience, let $S_{-i^*}$ denote all $S$ not including buyer $i^* \in N^*(\mathbf{t})$.

**Base Step:** For any $S \in S_{-1^*}$, it is clear that $W(S) \leq SW^*(\mathbf{t}_{-1^*})$. According to "2)", we know that

$$SW^*(\mathbf{t}_{-1^*}) \leq SW^*(\mathbf{t}_{-2^*}) - \hat{u}_{1^*} \leq SW^*(\mathbf{t}) - \sum_{i^* \in N^*(\mathbf{t})} \hat{u}_{i^*}. \tag{23}$$

According to (20) and the fact that $\hat{u}_i \geq 0$, we have that

$$W(S) \leq SW^*(\mathbf{t}_{-1^*}) \leq \hat{u}_0 \leq \sum_{i \in S} \hat{u}_i. \tag{24}$$

Therefore, any $S \in S_{-1^*}$ cannot be a blocking coalition.

**Induction Hypothesis:** Suppose $S$ cannot be a blocking coalition for all $S \in S_{-k^*}$ where $k = 1, 2, \cdots, i$.

**Induction Step:** For any $S \in S_{-i^*+1}$, we show that $S$ cannot be a blocking coalition. According to the induction hypothesis, we only need to consider all $S$ that includes $\{1^*, 2^*, \cdots, i^*\}$. For any $S \in S_{-i^*+1}$, we have that

$$W(S) \leq SW^*(\mathbf{t}_{-i^*+1}) \leq SW^*(\mathbf{t}_{-i^*+2}) - \hat{u}_{i^*+1} \tag{25}$$

$$\leq SW^*(\mathbf{t}) - \sum_{j^*=i^*+1}^{p^*} \hat{u}_{j^*} = \hat{u}_0 + \sum_{j^*=1^*}^{i^*} \hat{u}_{j^*} \leq \sum_{i \in S} \hat{u}_i. \tag{26}$$

Therefore, we have that for all $i^* \in N^*(\mathbf{t})$ and all $S \in S_{-i^*}$, $S$ cannot be a blocking coalition. Finally, consider all $S$ that includes the seller and $N^*(\mathbf{t})$, then according to "1)" and "3)", we have that

$$W(S) \leq W(N_0) = SW^*(\mathbf{t}) = \sum_{i \in N_0} \hat{u}_i = \sum_{i \in N^*(\mathbf{t}) \cup \{0\}} \hat{u}_i = \sum_{i \in S} \hat{u}_i. \tag{27}$$

Based on the above analysis, we conclude that for any type profile $\mathbf{t}$, the utility profile $\hat{u}$ is in the core. □

PROOF OF COROLLARY 4.3. Based on "1)" and "3)" of Proposition 2, we know that

$$\hat{u}_0 = SW^*(\mathbf{t}) - \sum_{i^* \in N^*(\mathbf{t})} \hat{u}_{i^*}. \tag{28}$$

According to "2)", we further have that $\sum_{i^* \in N^*(\mathbf{t})} \hat{u}_{i^*}$ is at most

$$\sum_{i^* \in N^*(\mathbf{t})} SW^*(\mathbf{t}_{-i^*+1}) - SW^*(\mathbf{t}_{-i^*}) = SW^*(\mathbf{t}) - SW^*(\mathbf{t}_{-1^*}), \tag{29}$$

where recall that $1^*$ is the first buyer in $N^*(\mathbf{t})$. Therefore, based on (28) and (29), we have that $\hat{u}_0 \geq SW^*(\mathbf{t}_{-1^*})$ for any $\hat{u}$ defined in the Proposition 2. □

PROOF OF COROLLARY 4.4. For any type profile $\mathbf{t}$, $CoreRev(N_0, W)$ represents the minimum core revenue. Proposition 1 proves that any core revenue is at least $SW^*(\mathbf{t}_{-\pi^*})$. Proposition 2 and Corollary 4.3 show that there exist core outcomes in which $\hat{u}_0$ is exactly $SW^*(\mathbf{t}_{-1^*})$. Therefore, we have $SW^*(\mathbf{t}_{-\pi^*}) \leq CoreRev(N_0, W) \leq SW^*(\mathbf{t}_{-1^*})$. □

# B  OMITTED PROOF FOR SECTION 5

PROOF OF THEOREM 5.1. Given any instance of the DAA with $\delta$ satisfying monotonic and strategy independent, we show that all buyers' utilities are non-negative and are maximized by reporting their types truthfully. Given a reported type profile $\mathbf{t}'$, suppose $\pi^*(\mathbf{t}') = \{1, 2, \cdots, w, w+1, \cdots, m\}$ and buyer $i$ reports her true type $(v_i, r_i)$, where $w$ denotes the winner in the DAA and $m$ denotes the winner of the efficient transaction. Next, we prove that $i$ cannot improve her utility by misreporting her type. For convenience, we use $\tilde{\mathbf{t}}'$ to denote the type profile under which $i$ misreports her type to $(v'_i, r'_i)$.

**Case 1:** $i \notin \pi^*(\mathbf{t}')$. According to Algorithm 2, buyer $i$'s utility is zero. The only way for $i$ to change her utility is to report $(v'_i, r'_i)$ such that she becomes the winner of the efficient allocation. Now, her payment is at least $SW^*(\tilde{\mathbf{t}}'_{-i}) + C^*_i(\tilde{\mathbf{t}}')$ and her utility is at most $v_i - C^*_i(\tilde{\mathbf{t}}') - SW^*(\tilde{\mathbf{t}}'_{-i})$. As $i \notin \pi^*(\mathbf{t}')$ and $C^*_i(\mathbf{t}')$ is not affected by $i$'s report, we have that $C^*_i(\tilde{\mathbf{t}}') = C^*_i(\mathbf{t}')$ and $SW^*(\tilde{\mathbf{t}}'_{-i}) = SW^*(\mathbf{t}')$, therefore $i$'s utility is at most $v_i - C^*_i(\mathbf{t}') - SW^*(\mathbf{t}') = SW^*_i(\mathbf{t}') - SW^*(\mathbf{t}') \leq 0$. That is, all buyers out of $\pi^*(\mathbf{t}')$ cannot increase utilities by misreporting.

**Case 2:** $i \in \pi^*_w(\mathbf{t}') \setminus \{w\}$. According to Algorithm 2, $i$'s utility is identical to $SW^*(\mathbf{t}'_{-\delta_i}) - SW^*(\mathbf{t}'_{-i})$ which is non-negative as $V(\mathbf{t}'_{-i}) \subseteq V(\mathbf{t}'_{-\delta_i})$. Since $i$ does not win, we know that $SW^*(\mathbf{t}'_{-\delta_i}) > SW^*_i(\mathbf{t}'_{-\delta_i})$. If buyer $i$ misreports to become the winner, her utility is at most $SW^*_i(\tilde{\mathbf{t}}'_{-\delta_i}) - SW^*(\tilde{\mathbf{t}}'_{-i}) = SW^*_i(\mathbf{t}'_{-\delta_i}) - SW^*(\mathbf{t}'_{-i}) < SW^*(\mathbf{t}'_{-\delta_i}) - SW^*(\mathbf{t}'_{-i})$. Therefore, it is no good for $i$ to become the winner. Moreover, suppose buyer $i$ is still a buyer in $\pi^*_w(\tilde{\mathbf{t}}') \setminus \{\tilde{w}\}$ after misreporting, then according to the MN property of $\delta_i$, we have that $SW^*(\mathbf{t}'_{-\delta_i}) \geq SW^*(\tilde{\mathbf{t}}'_{-\delta_i})$ and hence $i$'s utility is reduced. In sum, all buyers in $\pi^*_w(\mathbf{t}') \setminus \{w\}$ cannot increase their utilities by misreporting.

**Case 3:** $i = w$. Consider the following two subcases.

[i.] $CO^*_w(\mathbf{t}') = \emptyset$. Based on Algorithm 2, we know that the winner's payment is exactly $SW^*(\mathbf{t}'_{-i}) + C^*_i(\mathbf{t}')$ and her utility is identical to $SW^*(\mathbf{t}'_{-\delta_i}) - SW^*(\mathbf{t}'_{-i})$. On the one hand, if $i$ still wins after misreporting, then her payment is at least $SW^*(\tilde{\mathbf{t}}'_{-i}) = SW^*(\mathbf{t}'_{-i})$ and her utility is at most $SW^*(\mathbf{t}'_{-\delta_i}) - SW^*(\mathbf{t}'_{-i})$ (note that $CO^*_w(\tilde{\mathbf{t}}')$ may not be empty when $i$ misreports). On the other hand, if $i$ misreports to a buyer in Case 2, her utility becomes $SW^*(\tilde{\mathbf{t}}'_{-\delta_i}) - SW^*(\tilde{\mathbf{t}}'_{-i})$. Since $i$ wins in $\mathbf{t}'_{-\delta_i}$, then according to the

MN and SI properties, we have that $\text{SW}^*(\tilde{\mathbf{t}}'_{-\delta_i}) < \text{SW}^*(\mathbf{t}'_{-\delta_i})$. As $\text{SW}^*(\tilde{\mathbf{t}}'_{-i}) = \text{SW}^*(\mathbf{t}'_{-i})$, we know that $i$ cannot increase her utility. Therefore, the winner cannot increase her utility by misreporting when there is no critical opponent.

[ii.] $\text{CO}^*_w(\mathbf{t}') \neq \emptyset$. According to Proposition 4, the winner must be buyer $m$ and her utility can be expressed as

$$v_m - \max\{\text{SW}^*(\mathbf{t}'_{-m}) + C^*_m(\mathbf{t}'), \max_{k \in \text{CO}^*_m(\mathbf{t}')}\{\text{SW}^*_k(\mathbf{t}'_{-\delta_k}) + C^*_m(\mathbf{t}'_{-\delta_k})\}\}.$$

We first show that $m$ cannot be a buyer in Case 2 by misreporting. Consider the contrary that $m$ becomes a buyer in Case 2 by $(v'_m, r'_m)$. Then $\pi^*_m(\mathbf{t}') = \pi^*_{\tilde{m}}(\tilde{\mathbf{t}}') \subset \pi^*_m(\tilde{\mathbf{t}}')$, where $\tilde{m}$ is the winner in $\pi^*(\tilde{\mathbf{t}}')$. Now, consider any $k \in \text{CO}^*_m(\mathbf{t}')$. As $k$ is a critical opponent, we have that $m, \tilde{m} \in V(\mathbf{t}'_{-\delta_k})$ and $\pi^*_k(\mathbf{t}'_{-\delta_k})$ are the transaction with the second highest social welfare in $\mathbf{t}'_{-\delta_k}$. Therefore, we have that $\text{SW}^*_k(\mathbf{t}'_{-\delta_k}) \geq \text{SW}^*_{\tilde{m}}(\mathbf{t}'_{-\delta_k})$. When $m$ misreports, $\tilde{m}$ becomes the new winner of the efficient transaction, and thereby we must have that $\text{SW}^*_{\tilde{m}}(\tilde{\mathbf{t}}') > \text{SW}^*_m(\tilde{\mathbf{t}}')$. Based on the SI property and the fact that $m, \tilde{m} \in \tilde{\mathbf{t}}'_{-\delta_k}$, we further have that $\text{SW}^*_{\tilde{m}}(\tilde{\mathbf{t}}'_{-\delta_k}) > \text{SW}^*_m(\tilde{\mathbf{t}}'_{-\delta_k})$. In addition, it is clear that $\text{SW}^*_k(\tilde{\mathbf{t}}'_{-\delta_k}) \geq \text{SW}^*_{\tilde{m}}(\tilde{\mathbf{t}}'_{-\delta_k})$, now $\pi^*_k(\tilde{\mathbf{t}}'_{-\delta_k})$ becomes the transaction with the highest social welfare in $\tilde{\mathbf{t}}'_{-\delta_k}$! According to Algorithm 2, buyer $k$ has priority to win in advance, and thus $m$ cannot be a buyer of Case 2, which contradicts the assumption. Next, suppose $m$ is the winner in $\pi^*(\tilde{\mathbf{t}}')$ and still wins in Algorithm 2. For any buyer $k \in \text{CO}^*_m(\mathbf{t}')$, based on the SI property and Definition 11, we know that $k$ is still a critical opponent in $\tilde{\mathbf{t}}'$. However, under $(v'_m, r'_m)$ some other buyers may become new critical opponents, i.e., $\text{CO}^*_m(\mathbf{t}') \subseteq \text{CO}^*_m(\tilde{\mathbf{t}}')$, which will potentially increase $m$'s payment. In sum, $m$ cannot increase her utility by misreporting if there are critical opponents.

**Case 4:** $i \in \pi^*_m(\mathbf{t}') \setminus \pi^*_w(\mathbf{t}')$. According to Algorithm 2, $i$'s utility is zero. We first prove that $i$ cannot become a buyer in Case 2 by misreporting. Suppose $i$ becomes a buyer of Case 2 by submitting $(v'_i, r'_i)$, then $\text{SW}^*_i(\tilde{\mathbf{t}}') < \text{SW}^*_{\tilde{m}}(\tilde{\mathbf{t}}')$. As $\text{SW}^*_w(\mathbf{t}'_{-\delta_w})$ is the transaction with the highest social welfare in $\mathbf{t}'_{-\delta_w}$, it is clear that $\text{SW}^*_w(\mathbf{t}'_{-\delta_w}) \geq \text{SW}^*_{\tilde{m}}(\mathbf{t}'_{-\delta_w})$. Based on the SI property and the fact of $r'_i \subseteq r_i$, we further have that $\text{SW}^*_w(\tilde{\mathbf{t}}'_{-\delta_w}) \geq \text{SW}^*_{\tilde{m}}(\tilde{\mathbf{t}}'_{-\delta_w}) > \text{SW}^*_i(\tilde{\mathbf{t}}'_{-\delta_w})$. Therefore, $\text{SW}^*_w(\tilde{\mathbf{t}}'_{-\delta_w})$ is still the transaction with the highest social welfare in $\tilde{\mathbf{t}}'_{-\delta_w}$. According to Algorithm 2, $i$ cannot be a buyer of Case 2. Thus, the only way for $i$ to change her utility is to submit a report $(v'_i, r'_i)$ such that she becomes the winner of $\pi^*(\tilde{\mathbf{t}}')$ and also wins in Algorithm 2. In this case, she must beat $w$ in $\tilde{\mathbf{t}}'_{-\delta_w}$. Based on the SI property, we know that $\text{SW}^*_w(\tilde{\mathbf{t}}'_{-\delta_w})$ will become the transaction with the second highest social welfare in $\tilde{\mathbf{t}}'_{-\delta_w}$. According to Definition 11, $w$ will become a critical opponent of $i$! Now, $i$'s payment is at least $\text{SW}^*_w(\tilde{\mathbf{t}}'_{-\delta_w}) + C^*_i(\tilde{\mathbf{t}}'_{-\delta_w})$ and her utility is at most $v_i - C^*_i(\tilde{\mathbf{t}}'_{-\delta_w}) - \text{SW}^*_w(\tilde{\mathbf{t}}'_{-\delta_w})$. Recall that $C^*_i(\tilde{\mathbf{t}}'_{-\delta_w}) = C^*_i(\mathbf{t}'_{-\delta_w})$ and $\text{SW}^*_w(\tilde{\mathbf{t}}'_{-\delta_w}) = \text{SW}^*_w(\mathbf{t}'_{-\delta_w})$, so $i$'s utility is at most $\text{SW}^*_i(\mathbf{t}'_{-\delta_w}) - \text{SW}^*_w(\mathbf{t}'_{-\delta_w}) < 0$. In sum, all buyers $\pi^*_m(\mathbf{t}') \setminus \pi^*_w(\mathbf{t}')$ cannot increase their utilities by misreporting.

Based on the above analysis, we conclude that if $\delta$ is MN and SI, then the DAA defined in Algorithm 2 is IR and IC. $\square$

