# OpenReview forum: "Core-Competitiveness in Partially Observable Networked Market"
_ACM.org/TheWebConf/2024/Conference — TheWebConf24 Oral_

### Official Review · Reviewer_TwHv · 2023-11-03

**Novelty:** 6
**Technical Quality:** 7

**Review:**

### Summary
The authors consider partially observable networked markets and generalize the design of core-competitive auctions to this market setting. here, a mechanism consists of allocation and payment policies, where the allocation policy maps buyers' reported types to a directed path, aka a sequence of transactions from the seller to the final winning buyer, and the payment policy maps the reported types to buyers' payments. Typical notions such as IC, IR, non-wastefulness, and efficiency are defined in the same way.

The authors then define the core of a coalitional game based on $N_0$ (all sellers plus the buyer) and a coalitional value function. Then, they define the notion of core-competitive mechanisms, the minimum core revenue, and provide upper and lower bounds on the MCR. In doing so, the authors propose and characterize the set of critical buyers, and provide bounds on MCR based on the critical buyer set.

Then, the authors show that no auction mechanism can always generate core outcomes, indicating the incompatibility between efficiency and core-competitivenes. The authors then focus on designing mechanisms that are NW, IC, IR, CC, but not perfectly efficient. To this end, they introduce the notions such as link function profiles $\delta$ and critical opponents, and propose a deferred allocation auction algorithm (DAA) which takes a link function profile of all buyers and their reported types. They show that DAA is NW and furthermore IR and IC if $\delta$ is MN and SI, while the revenue can still be low. Then, they provide additional assumptions on $\delta$ (MN, SI, and PB) which ensure that DAA is IR, IC, and also CC. Furthermore, under the same assumptions on $\delta$, when the auction outcome is efficient, it must be in the core.

### Overall evaluation
New market setting with comprehensive results.

**Questions:**

Quick clarification questions:
- Line 226: fixed and known (to whom? Seller or buyers?)
- Line 227: Aka a directed acyclic path from the seller to the winning buyer?
- Similarly, in Definition 2, the allocation policy maps a set of reported buyer types to either a directed acyclic path from the seller to the winning buyer, which only contains valid buyers/arcs, or nothing (aka not selling it)?
- Line 381: Should use $\mathbf{t}'_S$ and $t'_i$ here and subsequently to denote reported types? Maybe explain that the prime $'$ will be dropped here to simplify notation. Might also mention that it can be dropped altogether by revelation principle (Line 569).

Questions:
- Definition 9: Is it possible that there are multiple SW-maximizing transactions (with multiple possible winners) for the same $\mathbf{t}$? If so, how would the definition handle it? Maybe a global lexicographical tie-breaking rule is needed to always ensure a unique winner? In other words, if multiple SW-maximizing transactions exist, then $\pi^*$ always picks the one that is lexicographically smaller than the other, and determines the unique winner accordingly.
- Since the motivation (Line 594) of developing DAA based on link function profiles is to provide tradeoff between efficiency and revenue, is there a (MN and SI?) link function profile that ensures allocation efficiency but sacrifices core-competitiveness? Examples 5.5 and 5.6 are ones that are MN, SI, and PB, which ensure CC but don't ensure efficiency.

nit
- Line 335: For technique convenience ---> Without loss of generality?
- Line 413: the seller [gets/obtains/achieves] a revenue that is...
- Line 414: In the following contents, ---> Next,

**Ethics Review Description:**

N.A.

**Reviewer Confidence:**

3: The reviewer is confident but not certain that the evaluation is correct

**Scope:**

4: The work is relevant to the Web and to the track, and is of broad interest to the community

---

### Official Review · Reviewer_UVJF · 2023-11-18

**Novelty:** 5
**Technical Quality:** 5

**Review:**

Summary:
The paper addresses core-competitive auctions in the context of Partially Observable Networked Markets (PONM), which captures real-world transaction markets where economic entities have limited information about others due to the market's vast size and exclusive information holdings. The paper defines PONM, studies auction design for such markets, and aims to generate revenue comparable to the core revenue while ensuring efficiency and stability. The work establishes upper and lower bounds for minimum core revenue in PONM and proves the impossibility of truthful, efficient, and core-competitive auctions. The paper identifies criteria for allocation rules and proposes a class of truthful auction mechanisms for PONM that are individually rational, incentive-compatible, and core-competitive.

Strengths: The introduction clearly defines the problem of information discrepancy in markets, emphasizing the limited knowledge of economic entities and the need for strategic disclosure of private information. This sets the stage for the exploration of PONM and core-competitive auctions.

Weaknesses:
1. Theorem 4.5 seems straightforward.
2. The proposed Deferred allocation auction only works well under a very specific setting.

**Questions:**

How do the proposed auction mechanisms account for practical considerations such as strategic behavior of economic entities or potential collusion?

**Reviewer Confidence:**

2: The reviewer is willing to defend the evaluation, but it is likely that the reviewer did not understand parts of the paper

**Scope:**

4: The work is relevant to the Web and to the track, and is of broad interest to the community

---

### Official Review · Reviewer_7g1x · 2023-11-20

**Novelty:** 5
**Technical Quality:** 5

**Review:**

This paper considers auction design within a graph setting, wherein a seller is connected and informed about a subset of buyers, who in turn are connected to others (under a graph structure) and may communicate (with cost) with them. The seller is looking to allocation and payment rules to sell a single item. This can be expressed as a directed path in the graph, which the terminal node being the allocation winner and costs incurred along the process. The seller's goal is to design an IC, IR mechanism whose revenue is core-competitive. That is, the revenue achieved is at least as much as a core satisfying allocation. The work is theoretical and (1) outlines a general model for graph structured auctions called Partially Observed Networked Market (PONM), (2) bound the core-satisfying revenue possible in such a market and (3) give constructive algorithms to find core-competitive allocations.

Overall, I think the model proposed in this work is novel and interesting. The authors do a good job of exploring the theoretical properties of core within this setting, including a known incompatibility between core and efficiency known in other auction settings. While I agree that the core is desirable notion and think the results in section 4 are meaningful, I am not sure if core-competitiveness is a strong desideratum. I expand on this and other concerns below. The writing itself, I find it a bit hard to parse something and think more can be done to improve readability.

**Questions:**

Could you provide a worked example of an allocation and payment rule for PONM setting. Is it correct to say that payment for any node who is not the winner must be 0 or -ve for this to be IR?

What is the time complexity of the proposed algorithm? It seems that for every buyer i, we compute the link function, recompute efficient/social welfare maximizing allocation with those links gone. This looks expensive? Also, what is the algorithm to compute the efficient allocation in PONM?

I am a bit confused by the definition of social welfare here (line 330). Traditionally, this should just be sum of all buyer utility - but here it seems to just be the value of the winner (not utility) minus all network cost. Isn't network/edge cost being covered by the platform/seller. Where is the payment paid by all buyers here in this social welfare.

I am a bit puzzled by the core-competitive notion. So this is an IC, IR outcome that is NOT in the core but achieves revenue equal to that of a core allocation. This may make sense as a comparison benchmark but in terms of desiderata, why not set the objective to just maximize revenue subject to IC and IR violation? Clearly if a core-competitive solution exists, this should find it (or something even better).

What does core-competitiveness mean when the core is empty? Analogously, how should I reconcile theorem 4.5 with the core-competitiveness results in section 5.

**Ethics Review Description:**

No ethics issue

**Reviewer Confidence:**

2: The reviewer is willing to defend the evaluation, but it is likely that the reviewer did not understand parts of the paper

**Scope:**

3: The work is somewhat relevant to the Web and to the track, and is of narrow interest to a sub-community

---

### Official Review · Reviewer_uRDL · 2023-11-21

**Novelty:** 5
**Technical Quality:** 6

**Review:**

Summary: The paper studies the design of core-competitive auctions for partially observable networked markets. These auctions are ones in which the seller's revenue is at least the least revenue it attains in any core outcome of the cooperative resource allocation game. The resource allocation problem is simple because there is a single item to allocate. Still, complexities arise since agents lie on a network with privately known linkages that can be used for item transfer, and the seller may not possess a direct link to each agent. The paper shows that while there is no auction mechanism that always guarantees a core outcome, core competitiveness can be achieved in an incentive-compatible and individually rational auction by sacrificing efficiency.

Strengths:
1. Very well-written paper
2. Novel analysis and strong results

Weaknesses:
1. Lack of practical motivation for the PONM model, especially important for an applied venue such as the WC.
2. Lack of insights into general properties of auctions necessary to achieve core competitiveness. In general, examples would help in illustrating why core-competitiveness can fail (e.g., under VCG) and how the new mechanism addresses the concern.

Post-rebuttal: Thanks for your convincing responses. I am happy to support the acceptance of the paper.

**Questions:**

Could you provide practical applications of the PONM model and the auction design objective?

**Reviewer Confidence:**

3: The reviewer is confident but not certain that the evaluation is correct

**Scope:**

3: The work is somewhat relevant to the Web and to the track, and is of narrow interest to a sub-community

---

### Official Review · Reviewer_sNqT · 2023-11-26

**Novelty:** 5
**Technical Quality:** 5

**Review:**

Summary:

This paper studies the problem of core-competitive auction design on a partially observable network. In this network market, each participant (a node in the network) can report their values and also their neighbors. The authors quantify the upper and lower bounds of the minimum core revenue and prove an impossibility result that there does not exist any truthful auction for PONM which is efficient and core-competitive. Moreover, the authors propose a new class of auction mechanisms for PONM that is individually rational, incentive-compatible, and core-competitive.

Comments:

This paper is generally well-written and clear. The problem studied in this paper is interesting and relevant to the auction design on social network. The collection of results is non-trivial and technically strong.

**Questions:**

1. Is it possible to provide approximation guarantee on the efficiency performance of the DAA mechanism against the best core revenue?

2. It seems that the property that the auction produces a core outcome whenever it generates an efficient allocation is a bit weak. Is it possible to obtain a mechanism that generates core outcomes for most instances?

**Reviewer Confidence:**

2: The reviewer is willing to defend the evaluation, but it is likely that the reviewer did not understand parts of the paper

**Scope:**

3: The work is somewhat relevant to the Web and to the track, and is of narrow interest to a sub-community

---

### Decision · Program_Chairs · 2024-01-22

**Decision:**

Accept (Oral)

**Comment:**

The paper studies a clean and novel problem in the "auction design in a social network" space, and executes quite well on it. The writing is clear, and the results are technically solid. It isn't completely clear how well motivated the problem is; the authors provided reasonable answers to this concern voiced in some reviews. As a package, the paper does reasonably well in all relevant dimensions, and could spark further study on this problem. Overall, this is a good fit for TheWebConf.